# Supply network position, digital transformation and innovation performance: Evidence from listed Chinese manufacturing firms

Chunyan Du[1,2]*, Qiang Zhang[1]

**1** Business School, Yunnan University of Finance and Economics, Kunming, Yunnan, China, **2** Financial Planning Department, China West Normal University, Nanchong, Sichuan, China

* candy4022@hotmail.com

**Data Availability Statement:** All relevant data are within the paper and its Supporting information files.

## Abstract

This study provides evidence for the supply network position influencing innovation performance and the moderating effect of digital transformation. Supply chain relationships have been evaluated in earlier research to demonstrate how concentrations of customers and suppliers may either favorably or adversely impact innovation. These metrics, however, only take into account how closely a firm is connected to its direct customers or suppliers. This study integrates the top five suppliers and customers of Chinese listed manufacturing firms and considers the relationship embeddedness of each firm's direct customers and suppliers, as well as the structure embeddedness among the customers' customers, customers' suppliers, suppliers' customers, and suppliers' suppliers to reveal the true impact of supply chain relationships on innovation performance. The top five suppliers and consumers of each firm are chosen to build a supply network for each year using panel data of listed Chinese manufacturing firms from 2013 to 2020. Social network analysis is used to determine network centrality and structural holes. The results show that in the supply network, network centrality and structural holes are significantly negatively correlated with innovation performance, especially in small and medium-sized firms, non-state-owned firms, and firms in recession phase. According to the moderating effect model, digital transformation is an efficient way to reduce the negative effect of supply network position on innovation performance. The research results will further improve the supply network cooperation mechanism, which is of great significance for improving supply chain resilience and firms' innovation.

## Introduction

A key factor in regional economic development and a crucial component of China's competitiveness in science and technology is the high-quality innovation of firms. Innovation performance is the competitive advantage that a firm obtains in the process of technological

**Funding:** The authors received no specific funding for this work.

**Competing interests:** The authors have declared that no competing interests exist.

innovation through R&D investment [1]. In fact, firms fully utilize external information and recognize the importance of knowledge transfer in improving innovation performance [2, 3]. Resource dependency theory stresses that organizations must connect with other organizations to receive essential external resources, which creates the dependence on external resources, and that they cannot only rely on internal resources for existence. Organizations can access and sustain external resources through cooperative partnerships with other organizations when faced with the unpredictability of resource acquisition and organizational reliance [4]. Firms in the market are widely networked and influence their related decisions by exchanging information and knowledge. Supply chain relationships bring opportunities for sustainable innovation within firms and originate from frequent contact between suppliers and customers. Choi et al. (2001) first proposed that the supply chain relationship should be regarded as a complex network of connections, and suppliers and customers should be regarded as actors in the network [5]. Innovation can be realized through the external network embedded with knowledge and information resources [6]. Suppliers and customers in the supply chain can offer external innovation resources that firms need. The sharing of knowledge and information between firms and supply chain partners is important for successful innovation [7].

The contributions of this study are as follows: (1) This study analyzes the effect of firms' network position on innovation performance by constructing a large sample of supply chain network. The findings of the study complement the existing literature and extend social network theory into supply chain research. (2) Existing research posits that firms in centrality and structural hole positions have information, resources, and control benefits that promote their profits [8, 9]. However, this study conducts an empirical analysis which reveals that firms in supply network positions would level down innovation performance.

In this study, a research framework on the effect of supply network position on firms' innovation performance is established. The paper is structured as follows: Section 2 reviews the literature on innovation performance and supply networks at home and abroad; Section 3 presents the research hypotheses, model specification, and variable settings; Section 4 discusses the results from the basic regression model, which is followed by a moderating effect analysis; and finally, the main conclusions, limitations, and policy implications of this study are highlighted in Section 5.

## Literature review

### Supply chain concentration and firms' innovation

Most of the existing studies consider supplier and customer concentration as crucial factors affecting innovation performance [10]. Supplier concentration may affect firms' innovation activities through: (1) benefit encroachment; (2) relationship transaction risk; and (3) knowledge-technology advantage. Benefit encroachment happens as supplier concentration rises, giving suppliers more power and opportunity to negotiate and encroach on the profits of other firms. Dowlatshahi (1999) confirmed that fewer suppliers in the market usually had a dominant position [11]. When a firm relies on only a few suppliers, the supplier controls the price, which squeezes the firm's product quality and profit margin. When firms invest in relationship-specific assets to increase their stickiness with suppliers, transaction risk in the relationship may result. Williamson (2002) pointed out that the market value of the specific asset existed in the relationship transaction [12], and Chen et al. (2008) argued that the investment in the specific asset implied the long-term maintenance of the relationship, and if the other party defaulted, the value of the specific asset would drop significantly, and reinvestment or adjustment would cause added costs, which would drag down the operational performance

[13]. Therefore, the investment in innovation would be lowered to reduce the risk. Technology-knowledge advantage occurs when there is a large concentration of suppliers, making it simpler for both sides to develop strong cooperation relationships that result in a greater exchange of information and technical support. This can bridge the technological gap between suppliers and customers [14].

Customer concentration acts on innovation activities from information feedback. Cool et al. (2010) argued that new needs and suggestions from customers were the starting point for new product R&D [15]. Manso (2011) stated that the key to driving innovation in firms was timely and effective information feedback [16]. Lukas and Ferrell (2000), Chu et al. (2019) also found that higher customer concentration implied closer interaction and information feedback [17, 18]. Customer concentration drive innovation process also result from knowledge spillover. Peer technology exchange, and R&D depend on close supplier-customer relationships [19]. Chu et al. (2019) examined whether supplier R&D innovation was affected by customer knowledge spillover and found that the geographical distance between customers and suppliers did affect suppliers' innovation [18].

Many existing studies use supplier and customer concentration as a measure of supply chain relationships to prove that customer concentration positively [20–24] or negatively [25–27] affects innovation, or that supplier concentration negatively affects firms' innovation [20, 24, 28–31]. However, these two indicators only reflect the relation embeddedness between the firm and its direct suppliers or customers; they do not express the structure embeddedness between the firm and customers' customers, customers' suppliers, suppliers' customers, and suppliers' suppliers. The above research has examined the linear supply chain relationship but failed to build the network-level supply chain relationship; therefore, the effect of supply chain relationships on innovation performance is not revealed.

## Supply network and firms' development

The embeddedness theory posits that economic behavior is embedded in the social structure [32], and the core of a social structure is the social network. Thus, the activities and decision-making of firms are inevitably affected by the social network in which they are positioned. As one of the branches of the social network, the supply chain network has a profound impact on firms' development.

Case study and questionnaire: Prior research has primarily used case studies or questionnaire surveys pertaining to the supply chain network. Choi and Hong (2002) characterized the supply network of the US auto industry in a case study [33]. Jian et al.(2013) found that relation embeddedness in the supply chain could improve innovation performance, while the structure embeddedness had nothing to do with innovation through questionnaires from 255 high-tech firms in Pearl River Delta [34]. Dong et al. (2015) studied the impact of relation and structure embeddedness on firms' opportunistic behavior by constructing a supply network in China's auto industry with the help of questionnaires and interviews [6]. Li et al. (2015) conducted a questionnaire survey of 237 manufacturing firms and observed that suppliers in network positions significantly promoted innovation performance [35].

Empirical studies on social network analysis find that network structure affects relational value [36], which may be measured by social network analysis [37]. Social network analysis is the process of investigating social structures through the use of networks and graph theory. It characterizes networked structures in terms of nodes (actors or things within the network) and the ties, edges, or links (relationships or interactions) that connect them. Social network analysis is the application of network science to social networks, and graphs are convenient mathematical representations, considered as the basis of this study. Then, Bellamy et al. (2014)

used social network analysis to create measures for each supply network structural characteristic for a sample of 390 firms, and found that supply network accessibility positively associated with a firm's innovation output [38]. Kao et al. (2017) built a supply network by integrating the main customer information of US listed firms from 2005 to 2012 and identified social network analysis measures most closely relate to supply chain efficiency [39]. Shi et al. (2019) constructed a supply chain network based on the names of the top five suppliers and customers of China's listed manufacturing firms from 2013 to 2016 and found that centrality and structural hole positions negatively affected the firms' operational performance [40]. Li et al. (2020) utilized the data of the electronics manufacturing industry of 200 listed firms in China from 2013 to 2017 and found that the centrality and structural holes in supply network positioning is significantly positively correlated with innovation performance by negative binomial model [41]. Yu et al. (2022) also constructed a supply chain network using the names of the suppliers and customers of China's top five listed firms from 2004 to 2019 and found that listed firms in a centrality position promoted firms' competitiveness [42].

Even though some research reports the relationship between supply network characteristics and firms' innovation for a single sub-sector industry, these studies lack empirical analyses using large samples of data to reveal the true effect of supply chain network position on innovation performance. Therefore, this study explores the relationship between supply network position and innovation performance and discusses the moderating effect of digital transformation.

## Research design

### Research hypotheses

The behavior of actors in social and economic activities embedded in an interactive network is affected by the social network [32, 43]. The social network emphasizes its information, resource, and control benefits through relationship embeddedness and structure embeddedness. In the supply network, relationship embeddedness is manifested as repeated cooperation among network members, which can enhance trust and strengthen information sharing and resource complementarity. Existing studies have utilized customer and supplier concentrations to characterize relationship embeddedness [25, 44, 45]. We focus on the effect of structural embeddedness in the supply network on innovation performance. Structure embeddedness emphasizes the informational roles that members occupy in the overall network [46], reflecting the aspect of information sharing resulting from interactions among members [47, 48]. Network centrality and structural holes characterize the structure positions occupied by firms in a network. Therefore, we explore the effect of supply network centrality and structural holes on innovation performance.

**Centrality and innovation performance.**   Network centrality is frequently used to assess the status and social prestige of social network members [49]. The greater the network centrality of a firm, the greater the number of partners of that firm. Firms occupying a centrality position in a social network have access to certain resources, information, and control benefits [50]. As they conduct more direct or indirect business with other firms in the network, it becomes easier for them to strengthen the transfer of resources and acquire more advanced technology and knowledge, which contribute to improving innovation capabilities. These firms have many channels through which they can collect rich information to reduce transaction costs and enhance innovation capabilities [42]. Firms with higher centrality have more cooperation choices. This asymmetry of choice determines that firms with higher network centrality have stronger bargaining power in supply chain relationships [51, 52], thereby improving their own profitability and innovation capabilities [53].

H1: There is a positive correlation between network centrality and innovation performance in a supply network.

**Structural holes and innovation performance.** Let us assume that firms A, B, and C exist in a network. A has direct connections with B and C in the social network, while B has no connections with C. The missing link between B and C indicates that there is a hole in the network structure [54]. According to the structural hole theory, a firm occupying a "bridge" position in the supply network can build a platform for communication between firms without direct business relationships. The greater the number of disconnected firms it connects, the greater the number of structural holes. Firms that are not directly connected to the network belong to different circles; each circle has its own unique technology and product pricing information. Firms occupying "bridge" positions can obtain information from other circles. More structural holes in a supply network could help firms access more valuable, non-redundant resources from multiple partners [44] and thus grow diverse technology and knowledge. This would be an incentive for firms to invent new technological breakthroughs. Kim (2017) found that structural holes brought more heterogeneity and complementary knowledge to firms and were conducive to improving technological innovation [55]. Concurrently, firms in a structural hole position could have strong bargaining power in the transaction process through the information and control benefits in the supply network. Bargaining power not only determines firms' profits, which could reduce operational risks with the help of sufficient innovation funds, but also makes it possible to avoid yielding to the claims of major customers, such as reducing sales prices, offering more commercial credit [56], or reducing costs related to innovative activities.

H2: There is a positive correlation between structural hole position and innovation performance in a supply network.

**The moderating effect of digital transformation.** Digitalization includes traditional information technology as well as "blockchain", "cloud computing", "artificial intelligence", and "big data". Technology is the underlying support for the digital transformation. Digital transformation is the process of industrial upgrading and transformation using emerging technologies [57]. Digital transformation can transform a firm's original operating model and internal organization and improve its performance by enhancing its learning capabilities and helping it embed itself in external networks [58]. As for the supply network, digital transformation will optimize the rationale for supply chain structure by improving the sensitive response to the market. Through big data analysis, firms can quickly identify the dynamic and diverse needs of consumers and potential suppliers. In order to keep the resilience of the supply network, the firms would revise their positions and strengthen network structural adjustments to meet the value creation. Meanwhile, digital transformation reveals the diversity of the supply network. Digital technology speeds up the exchange of internal and external information [59], which stimulates more cooperation in the supply chain network. In the practice of digital transformation, the supply chain network has created cross-space virtual cooperation, effectively breaking the limitations of traditional supply chain integration. In addition, the application of advanced digital technologies helps monitor members in the supply network [60] and utilize their resources scientifically.

H3:Digital transformation moderates the relationship between supply network position and innovation performance.

## Sample selection and data resource

"Standards on the Contents and Formats of Information Disclosure by Firms Offering Securities to the Public No. 2-Contents and Formats of Annual Reports" was published by the China

Securities Regulatory Commission in 2012. It encourages listed firms to record the names of their top five suppliers and customers. Therefore, we construct the supply network for each year by collating the names of the top five suppliers and customers stated in the annual reports of each listed firm from the CSMAR database. The reason why suppliers and customers of the network node firm should not be separate is that each supplier or customer in the directed network is a node for knowledge and information transfer. After we eliminate the samples with missing information, data for 2981 firms is finally obtained. We construct the directed supply network with a one-mode matrix and calculate centrality and structural holes of each network node firm accordingly through social network analysis using the Gephi software.

## Variable setting

The explained variable, innovation performance (RD), can be measured by the logarithm of R&D investment to eliminate heteroskedasticity.

The explanatory variable, that is, the network position, can be expressed by the centrality and structural holes. The centrality of nodes in the network mainly includes degree centrality, betweenness centrality, eigenvector centrality, and closeness centrality. Degree centrality (DC) refers to the number of firms in the network that are directly connected, which reflects the degree of control over the resources of core firms in the network [61]. Betweenness centrality (BC) measures the role of a firm in acting as a "bridge" between other firms. Eigenvector centrality (EC) is measured by considering the degree centrality of other firms connected in the network. Closeness centrality (CC) measures the degree of independence of firms in obtaining information and resources in a connected network. Since the supply network is not fully connected, closeness centrality is not suitable for our analysis. This study adopts DC as the measurement scale for network centrality position.

Structural holes (SH) are utilized to measure the redundancy of resources in the network [51]. If two firms establish a connection with a third firm simultaneously, but there is no direct connection between the two firms, the third firm is considered to be in a structural hole position [62].

The moderating effect variable, digital transformation (DT), can be indicated by the percentage of digital technology intangible assets [57]. We select digital technology intangible assets by searching for specific keywords in the intangible assets ledger of listed manufacturing firms, like artificial intelligence, big data, blockchain, cloud computing, intelligent manufacturing, and intelligent platforms.

The following control variables are considered: Lev, Roa, Share, Grow, Fixed_as and Size. This study also controls for industry and year-fixed effects to avoid inter-industry differences and time trends in the regression results. The specific measures of each variable are shown in Table 1.

## Model specification

The following models are constructed:

Basic model:

$$\ln RD_{it} = \alpha_0 + \alpha_1 \ln DC_{it} + \alpha_2 control_{it} + \lambda_{ind} + \lambda_t + \varepsilon_{it} \tag{1}$$

$$\ln RD_{it} = \beta_0 + \beta_1 SH_{it} + \beta_2 control_{it} + \lambda_{ind} + \lambda_t + \varepsilon_{it} \tag{2}$$

Here, $RD_{it}$ refers to the innovation performance of the $i_{th}$ firm in year t; $DC_{it}$ and $SH_{it}$ is the network degree centrality and structural holes of the $i_{th}$ firm in the $t_{th}$ year; $control_{it}$ denotes a vector consisting of a series of control variables involving Lev, Roa, Fixed_as, Share, Grow and

**Table 1. Measurement of each variable.**

| Name | Symbol | Formulas/Explanation |
|---|---|---|
| Innovation performance | lnRD | Log(R&D investment) |
| Degree centrality | lnDC | $\log \sum_{i=1}^{n} x(p_i p_j)$, n is the number of network nodes. If nodes i and j connect, $x(p_i p_j) = 1$; otherwise, $x(p_i p_j) = 0$. |
| Structural holes | SH | $1 - \sum_{j}(p_{ij} + \sum_{q} p_{iq}p_{qj})^2$, $p_{ij}$ is the proportion of the direct connections between nodes i and j in all the connections of i in the network; $p_{iq}$ and $p_{qj}$ are the proportion of the indirect connections between nodes i and j in all the connections of i in the network. |
| Asset-liability ratio | Lev | Total liability/total assets |
| Return on asset | Roa | Net profit/average total assets |
| Proportion of the largest shareholders | Share | Number of shares held by the largest shareholder/total number of shares of the firm |
| Tobin's Q value | Grow | Market value of debt and equity/replacement value of assets |
| Proportion of fixed assets | Fixed_ass | Fixed assets/ total assets |
| Scale of the firm | lnSize | Log(total assets) |
| Digital transformation | DT | Intangible assets related to digital transformation/total intangible assets |

lnSize. $\lambda_{ind}$, $\lambda_t$ and $\varepsilon_{it}$ represent the industry-fixed effect, year-fixed effect and random disturbance respectively.

Moderating effect model:

$$\ln RD_{it} = \theta_0 + \theta_1 \ln DC_{it} + \theta_2 DT_{it} + \theta_3 \ln DC_{it} \cdot DT_{it} + \theta_4 control_{it} + \varepsilon_{it} \qquad (3)$$

$$\ln RD_{it} = \mu_0 + \mu_1 SH_{it} + \mu_2 DT_{it} + \mu_3 SH_{it} \cdot DT_{it} + \mu_4 control_{it} + \varepsilon_{it} \qquad (4)$$

Here, $DT_{it}$ represents digital transformation of the $i_{th}$ firm in year t. It is necessary to consider each model coefficient and its significance, as this aids in investigating the relationship between supply network position and innovation performance and the moderating effect of digital transformation.

## Empirical results and discussion

### Descriptive statistics and analysis

Descriptive statistics and spearman correlation analysis of variables are conducted in the supply network using the STATA software (Tables 2 and 3). A multicollinearity analysis is also performed; the results reveal that the maximum variance inflation factor (VIF) in Model (1)-(4) is 2.55, 2.54, 1.86 and 1.86, reflecting that there is no multicollinearity between the variables.

### Basic model test and its robustness

Table 4 shows the regression results for the effect of supply network position on innovation performance. As indicated, network centrality affects innovation performance negatively, and the coefficient of degree centrality is -0.136 ($p<0.01$) in Model (1). Structural holes affect innovation performance significantly at the 1% level, and the coefficient is still negative (-0.324) in Model (2). The results demonstrate that supply network position has a notably negative influence on innovation performance. The empirical results strongly disagree with H1 and H2. With an increase in centrality and structure holes, the firm does not obtain resources,

**Table 2. Descriptive statistics of each variable.**

| Variable | Obs | Mean | Std.Dev. | Min | Max |
|---|---|---|---|---|---|
| lnRD | 2,981 | 17.946 | 1.783 | 5.094 | 25.025 |
| lnDC | 2,981 | 1.647 | 0.906 | 0 | 2.708 |
| SH | 2,981 | 0.682 | 0.352 | 0 | 0.918 |
| Lev | 2,981 | 0.431 | 0.202 | 0.008 | 0.994 |
| Roa | 2,981 | 0.032 | 0.079 | -0.827 | 0.464 |
| Fixed_as | 2,981 | 0.239 | 0.148 | 0.001 | 0.808 |
| Share | 2,981 | 33.288 | 14.646 | 3.89 | 89.99 |
| Grow | 2,981 | 2.063 | 1.480 | 0.699 | 17.729 |
| lnSize | 2,981 | 22.285 | 1.422 | 18.349 | 28.637 |
| DT | 2,981 | 0.0409 | 0.107 | 0 | 1 |

**Table 3. Spearman correlation analysis of each variable.**

| Variable | lnRD | lnDC | SH | Lev | Roa | Fixed_as | Share | Grow | lnSize | DT |
|---|---|---|---|---|---|---|---|---|---|---|
| lnRD | 1.000 | | | | | | | | | |
| lnDC | -0.239*** | 1.000 | | | | | | | | |
| SH | -0.263*** | | 1.000 | | | | | | | |
| Lev | 0.339*** | -0.084*** | -0.121*** | 1.000 | | | | | | |
| Roa | 0.113*** | -0.13*** | -0.115*** | -0.412*** | 1.000 | | | | | |
| Fixed_as | 0.014 | -0.026 | -0.001 | 0.141*** | -0.151*** | 1.000 | | | | |
| Share | 0.151*** | -0.066*** | -0.076*** | 0.085*** | 0.076*** | 0.091*** | 1.000 | | | |
| Grow | -0.346*** | 0.071*** | 0.105*** | -0.392*** | 0.266*** | -0.174*** | -0.142*** | 1.000 | | |
| lnSize | 0.687*** | -0.203*** | -0.244*** | 0.571*** | 0.055*** | 0.099*** | 0.177*** | -0.564*** | 1.000 | |
| DT | 0.062*** | -0.071*** | -0.052** | -0.063*** | 0.102*** | -0.222*** | 0.004 | 0.071*** | -0.048*** | 1.000 |

* $p < 0.1$,

** $p < 0.05$,

*** $p < 0.01$

**Table 4. Results of basic model regression.**

| | Model(1) | | | Model(2) | | |
|---|---|---|---|---|---|---|
| | Coef. | SE | P-value | Coef. | SE | P-value |
| lnDC | -0.136*** | 0.025 | 0.000 | | | |
| SH | | | | -0.324*** | 0.064 | 0.000 |
| Lev | -0.544*** | 0.135 | 0.000 | -0.549*** | 0.135 | 0.000 |
| Roa | 1.614*** | 0.295 | 0.000 | 1.635*** | 0.295 | 0.000 |
| Fixed_as | 0.001 | 0.175 | 0.996 | 0.01 | 0.175 | 0.956 |
| Share | 0.006*** | 0.002 | 0.000 | 0.006*** | 0.002 | 0.000 |
| Grow | 0.005 | 0.016 | 0.737 | 0.006 | 0.016 | 0.712 |
| lnSize | 0.934*** | 0.022 | 0.000 | 0.937*** | 0.022 | 0.000 |
| _cons | -3.811*** | 0.610 | 0.000 | -3.850*** | 0.612 | 0.000 |
| Industry | Y | | | Y | | |
| Year | Y | | | Y | | |
| N | 2981 | | | 2981 | | |
| adj. R2 | 0.63 | | | 0.63 | | |

**Table 5. Results of basic model robustness.**

| | Model(1)-1 | | | Model(1)-2 | | | Model(2)-1 | | | Model(2)-2 | | |
|---|---|---|---|---|---|---|---|---|---|---|---|---|
| | Coef. | SE | P-value | Coef. | SE | P-value | Coef. | SE | P-value | Coef. | SE | P-value |
| lnDC | -0.296*** | 0.032 | 0.000 | | | | | | | | | |
| SH | | | | | | | -0.710*** | 0.083 | 0.000 | | | |
| EC | | | | -1.239*** | 0.310 | 0.000 | | | | | | |
| BC | | | | | | | | | | -0.004*** | 0.001 | 0.000 |
| Lev | -0.346** | 0.175 | 0.048 | -0.343* | 0.177 | 0.053 | -0.358** | 0.175 | 0.041 | -0.361** | 0.176 | 0.041 |
| Roa | 1.945*** | 0.381 | 0.000 | 2.344*** | 0.383 | 0.000 | 1.989*** | 0.382 | 0.000 | 2.254*** | 0.382 | 0.000 |
| Fixed_as | -0.151 | 0.225 | 0.502 | -0.079 | 0.228 | 0.728 | -0.133 | 0.226 | 0.557 | -0.07 | 0.227 | 0.758 |
| Share | 0.004** | 0.002 | 0.039 | 0.004** | 0.002 | 0.035 | 0.004** | 0.002 | 0.034 | 0.005** | 0.002 | 0.016 |
| Grow | 0.031 | 0.021 | 0.131 | 0.044** | 0.021 | 0.032 | 0.032 | 0.021 | 0.118 | 0.04* | 0.021 | 0.055 |
| lnSize | 0.555*** | 0.029 | 0.000 | 0.628*** | 0.029 | 0.000 | 0.561*** | 0.029 | 0.000 | 0.616*** | 0.028 | 0.000 |
| _cons | -9.626*** | 0.788 | 0.000 | -11.56*** | 0.778 | 0.000 | -9.705*** | 0.791 | 0.000 | -11.29*** | 0.773 | 0.000 |
| Industry | Y | | | Y | | | Y | | | Y | | |
| Year | Y | | | Y | | | Y | | | Y | | |
| N | 2981 | | | 2981 | | | 2981 | | | 2981 | | |
| adj. R2 | 0.375 | | | 0.360 | | | 0.372 | | | 0.365 | | |

information, and control benefits, which is unfavorable for innovation performance. The reason may be that the network position would benefit the firms based on the assumption that resource sharing and information transmission of each firm in the social network is absolutely effective. As far as the central position of the supply network is concerned, even if the firm can contact other firms by its core position, which has more channels for obtaining resources and information, a low degree of marketization makes it hard to facilitate the free flow of information and resources among firms. Although firms in structural holes can theoretically obtain more heterogeneous external information, information asymmetry among firms also makes it difficult to establish mutual trust and promote effective transmission, thus weakening the firm's innovation ability.

A robustness test is also performed to increase the credibility of the results. There are numerous methods for testing robustness, such as alternative variables, supplementary variables, sub-sample regression, model substitution, and changing the sample size. For the robustness test, we replace R&D investment, degree centrality, and structural holes in the model with the number of patent applications, eigenvector centrality [38], and betweenness centrality [63], respectively. The relevant empirical results are shown in Table 5. The direction of each variable is consistent with Model(1) and (2). The coefficients of degree centrality in Model(1)-1 and Model(1)-2 and structural holes in Model(1)-1 and Model(1)-2 are different from one another. Hence, the results of Model(1) and (2) are robust.

## Heterogeneity analysis of basic model

The effect of supply network position on innovation across various firm sizes is seen in Fig 1 The firms are categorized into large firms and small and medium-sized ones based on the mean value of the total assets. Small and medium-sized firms have weaker financial strength and fewer resources and control benefits than large firms. There is a greater likelihood that they may become entangled by the terms of the cooperative partners' operations, and the relationship inside the network could be full of higher risks. Therefore, the centrality of small and medium-sized firms has a greater detrimental effect on innovation performance than that of

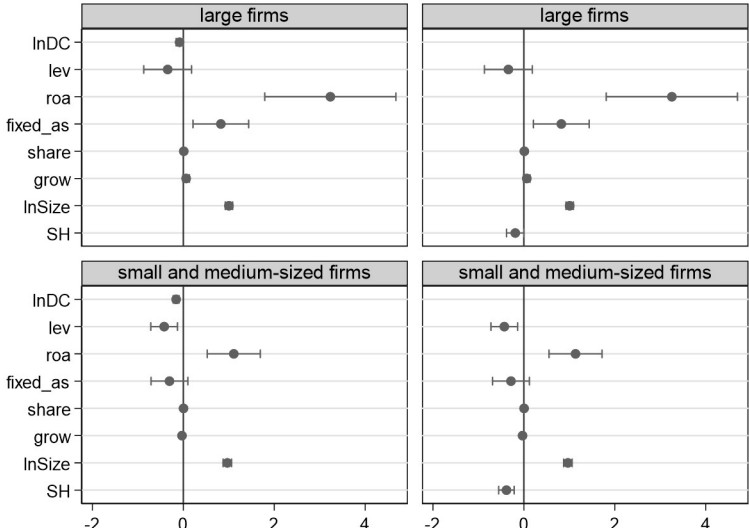

**Fig 1. The effect of supply network position on innovation under different firm sizes.**

large firms. Weisbuch and Battiston (2007) confirmed that partial failures in the supply network would lead to a rapid shortage of the entire networks' products, resulting in some firms going bankrupt in a simulation analysis [64]. The losses brought on by opportunistic behavior will also be bigger than those of large firms, and the more structural holes small and medium-sized firms have, the less they can control their partners. Therefore, compared to large firms, small and medium-sized firms are more sensitive to the performance of innovation due to network centrality and structural holes.

Fig 2 details the relationship between supply network position and innovation in state-owned and non-state-owned firms. State-owned firms may be more prominent, which may explain why the centrality and structural holes of non-state-owned firms have a bigger

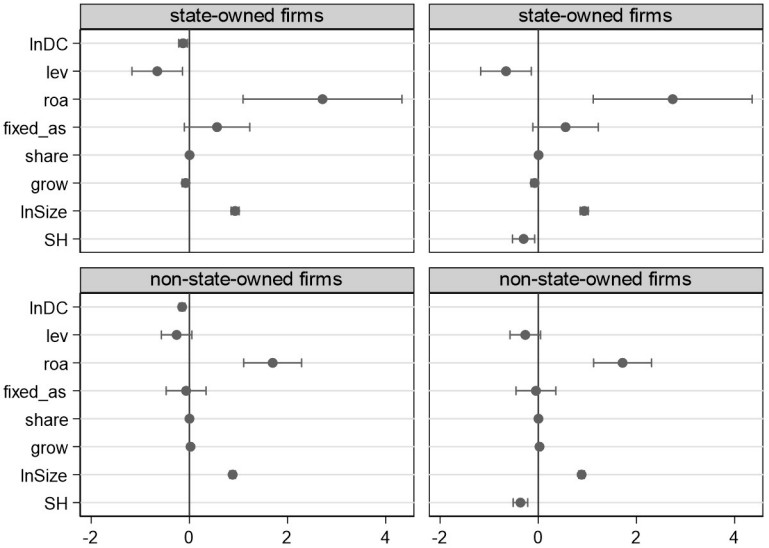

**Fig 2. The effect of supply network position on innovation in state-owned and non-state-owned firms.**

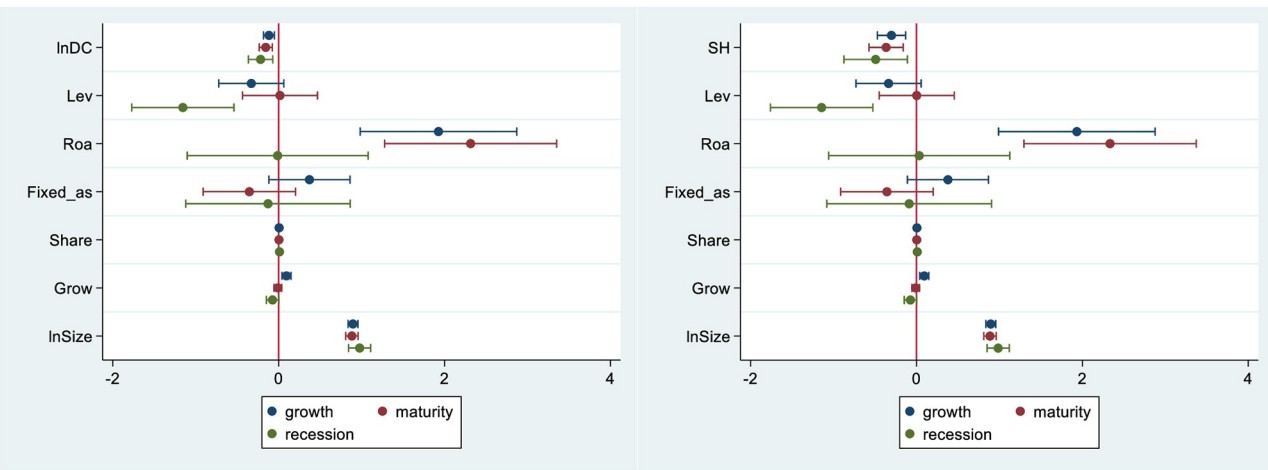

**Fig 3. The effect of supply network position on innovation at various stages.**

detrimental effect on innovation than state-owned firms. State authority safeguards the development of state-owned businesses despite the fact that their organizational inertia results in significant dependence and a constrained flow of resources. Costs, benefits, and network position do not determine how innovative state-owned firms are. Non-state-owned firms are more likely to be impacted by the supply network while having higher operational risk and financial costs. The numerous structural holes in non-state-owned firms make it impossible to effectively stop partners from behaving opportunistically, and the absence of a trust-building mechanism will lead to unstable cooperative partnerships. This hurts firms' ability to tap outside innovative resources.

The firm's life cycle can be separated into three phases according to net cash flows from operating activities, investing activities, and financing activities: growth, maturity, and recession [65]. Fig 3 displays the effect of supply network position on innovation at various stages. We observe that network position has a detrimental impact on innovation in all phases, with the effects during the recession being the most pronounced. This could be as a result of the fact that firms in the growth phase have high learning and absorptive skills to gather non-redundant and heterogeneous knowledge to drive expansion, whereas firms in the maturity phase have great resource accumulation. Some of the detrimental effects of operational risk brought on by network instability can be mitigated by the strong innovation pace in the two phases. On the other hand, firms in the recession phase have no resources and control advantages, and their funds are more used to maintain basic survival, so it is more difficult to find breakthroughs in innovation.

## Moderating effect model test

It can be seen that digital transformation significantly positively affects innovation performance significantly from Table 6, that is, a 1% rise in digital transformation can increase the firms' innovation approximately by 0.6%. Digital technology applications reduce the cost of innovation. For example, technologies such as the Internet of Things reduce information search and communication costs and achieve cross-regional allocation of innovation resources [66]. Digital technologies also enhance innovation performance through networked collaborative manufacturing. Networked collaborative manufacturing makes firms realize the collaboration between internal R&D and supply chain partners and the sharing of

**Table 6. Results of the moderating effect of digital transformation.**

| | Model(3) | | | Model(4) | | |
|---|---|---|---|---|---|---|
| | Coef. | SE | P-value | Coef. | SE | P-value |
| lnDC | -0.210*** | 0.027 | 0.000 | | | |
| SH | | | | -0.512*** | 0.069 | 0.000 |
| DT | 0.591*** | 0.218 | 0.007 | 0.604*** | 0.218 | 0.006 |
| lnDC*DT | 0.391* | 0.232 | 0.092 | | | |
| SH*DT | | | | 1.129* | 0.59 | 0.056 |
| Lev | -0.666*** | 0.146 | 0.000 | -0.675*** | 0.146 | 0.000 |
| Roa | 1.151*** | 0.317 | 0.000 | 1.167*** | 0.318 | 0.000 |
| Fixed_as | -0.788*** | 0.159 | 0.000 | -0.792*** | 0.159 | 0.000 |
| Share | 0.002 | 0.002 | 0.177 | 0.002 | 0.002 | 0.144 |
| Grow | -0.001 | 0.017 | 0.963 | -0.001 | 0.017 | 0.933 |
| lnSize | 0.886*** | 0.022 | 0.000 | 0.888*** | 0.022 | 0.000 |
| _cons | -1.103** | 0.472 | 0.019 | -1.145** | 0.474 | 0.016 |
| N | 2981 | | | 2981 | | |
| adj. R2 | 0.517 | | | 0.516 | | |

data resources in the supply chain, thus shortening the product R&D cycle and improving the innovation output.

The coefficient of digital transformation × degree centrality(0.391) and digital transformation × structural holes(1.129) are bigger than those of degree centrality (-0.210) and structural holes(-0.512) at the 10% level in Model(3) and Model(4), which reflect that digital transformation weakens the adverse effects of supply network position on innovation in the firms significantly. The empirical results strongly support H3. Digital transformation is an efficient way of moderating the impact of supply network position on innovation performance.

The use of digital technologies allows for further optimization and coupling of information collection, analysis, and processing as well as application within the firm. Fig 4 confirms that digital transformation negatively moderates the relationship between network position and innovation performance. The slope becomes smaller in firms with high digital transformation compared to those with low digital transformation. Because firms with high digital

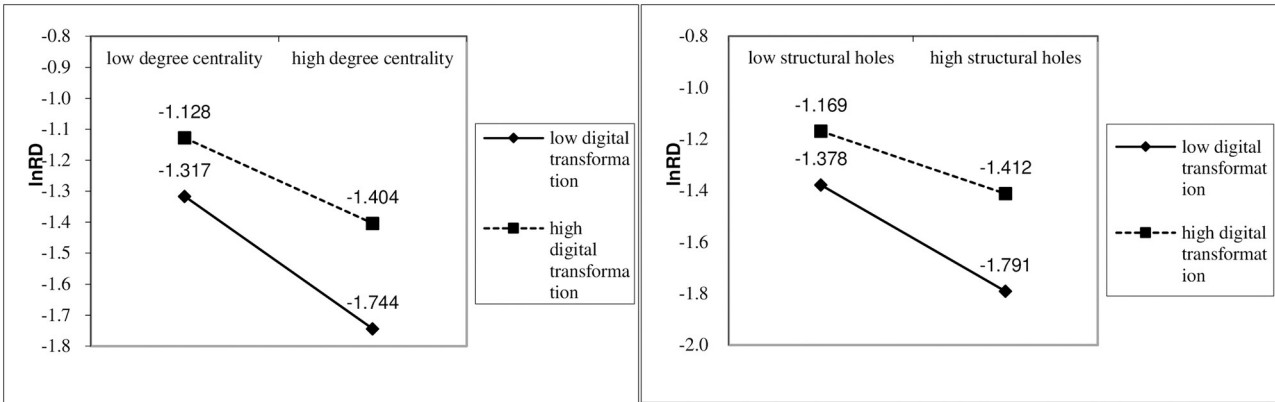

**Fig 4. The moderating effect of digital transformation.**

transformation help to improve the exchange of multilateral information with supply chain partners, reconfiguring their supply position, which reshapes innovation performance.

## Comparison with other studies

The cited research provides the theoretical basis for the analysis in this study. To begin, unlike previous studies, we combine social network analysis with supply chain analysis for our research. Although the academic research on the effectiveness of supplier and customer concentration has attracted extensive attention, the existing research does not focus on the network analysis of the firm perspective. Some studies posit that firms in network positions promote their development. However, this study finds that supply network position could negatively affect innovation performance. Second, a few empirical analyses highlight digital transformation advantages in the supply chain. In addition, this study has reference significance for the research on China's manufacturing industry, which is the backbone of the national economy.

## Conclusions and policy implications

This study integrates the data from the top five suppliers and customers of Chinese listed manufacturing firms and considers the relationship embeddedness of each firm's direct customers and suppliers, as well as the structure embeddedness among the customers' customers, customers' suppliers, suppliers' customers, and suppliers' suppliers. Through social network analysis, we construct a supply network of manufacturing firms with the data from 2013 to 2020 and measure the effect of supply network position on innovation performance. The research conclusions are as follows: First, the degree centrality and structural holes in the supply network did not promote innovation performance. Results suggest that in the supply network, the resources, information, and control advantages of degree centrality and structural holes have not been effectively exerted. This effect is more obvious in small and medium-sized firms, non-state-owned firms, and firms in the recession phase. Second, digital transformation is an efficient way to reduce the negative effect of supply network position on innovation performance.

   The findings also have some policy implications. First, firms should maintain a harmonious trading environment and actively create an efficient supply network. The supply network may lack resilience when one firm in the network is in crisis, and other firms are likely to be affected. This suggests that firms embedded in the same supply network should jointly maintain a harmonious and orderly trading environment, resist market opportunism, and promote mutual trust and cooperation with their partners to reduce risks. Second, optimizing external environmental benefits fosters innovation in a supply network. Firms located in the center of the supply network and occupying more structural holes face greater risks and financial constraints, reducing their willingness to innovate. Therefore, firms should stay in an adequate position in the supply network and appropriately replace the concentration of suppliers and customers. Moreover, firms should focus on establishing strong ties with other firms and establish strategic alliances with supply network members to govern vicious market competition. This prevents opportunistic behavior by partners and reduces risk. Third, firms should formulate long-term plans for digital strategies considering their own network position, making up for the shortage of resources through the construction of basic digital capabilities, paying special attention to combining digital strategies with the current market and technology.

   This study has limitations, which it is important for future research to address. First, there is scope for a follow-up study on the effect of supply network position on innovation performance in some other industries (such as energy and agriculture), which may have different

results and findings. Second, suppliers' and customers' information about non-listed firms could be adopted to further extend the established network. Third, the nonlinear relationship or mediation effect could be explored to understand more clearly how they interact with each other. All of these could create more breakthroughs for suggestions in the next step.

## Supporting information

**S1 Table. Regression data.**
(XLSX)

## Author Contributions

**Conceptualization:** Qiang Zhang.

**Data curation:** Chunyan Du.

**Formal analysis:** Chunyan Du.

**Methodology:** Chunyan Du.

**Project administration:** Qiang Zhang.

**Resources:** Chunyan Du.

**Supervision:** Qiang Zhang.

**Writing – original draft:** Chunyan Du.

**Writing – review & editing:** Chunyan Du, Qiang Zhang.

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
