## [Decision Letter · Decision Letter 0]

31 Aug 2022

PONE-D-22-18835Supply network position, digital transformation and innovation performance: evidence from listed Chinese manufacturing firmsPLOS ONE

Dear Dr. du,

Thank you for submitting your manuscript to PLOS ONE. After careful consideration, we feel that it has merit but does not fully meet PLOS ONE’s publication criteria as it currently stands. Therefore, we invite you to submit a revised version of the manuscript that addresses the points raised during the review process. Please address all requested corrections and amend your manuscript accordingly.

We look forward to receiving your revised manuscript.

Kind regards,

Ashkan Memari

Academic Editor

PLOS ONE

Journal Requirements:

Reviewers' comments:

Reviewer's Responses to Questions

**Comments to the Author**

1. Is the manuscript technically sound, and do the data support the conclusions?

Reviewer #1: Yes

Reviewer #2: Partly

2. Has the statistical analysis been performed appropriately and rigorously? 

Reviewer #1: Yes

Reviewer #2: Yes

3. Have the authors made all data underlying the findings in their manuscript fully available?

Reviewer #1: No

Reviewer #2: No

4. Is the manuscript presented in an intelligible fashion and written in standard English?

Reviewer #1: Yes

Reviewer #2: Yes

5. Review Comments to the Author

Reviewer #1: Dear Authors,

Overall, the paper was very interesting to read and presented an innovative approach to tackle your research questions. The results also have practical implications and imply a new contribution to research. In general, the paper was easy to follow, however there are some parts which need further major revision to make it more clear to the reader.

1. Introduction:

Innovation performance is a central focus of your paper, yet it is not clearly defined for the reader. You state that is is necessary for development and competitiveness, but not what is it or how it could be measured (this is mentioned much later in the methods section). Please clearly define this in the opening paragraph.

2. Literature review:

This section needs a major revision. It is unclear to read, and for some studies you discuss the methodology used, while in others the results of the study as well. Please break this into smaller sections which directly relate to the focus of this research. This could also be broken into sections by results (i.e. studies that found centrality (or some similar characteristic) positively correlated with innovation, and those that found it negatively). You may also want to indicate which are studies based in China, and which are abroad (or a mix of international studies), so that you can discuss the generalizability to other countries/regions.

The reader also does not understand clearly what gap you are filling. From reading the paper it appears the large dataset, as well as the incorporation of the time aspect is where the major contribution lies because you indicate that some other studies also found that centrality may negatively impact innovation. Please also discuss social network theory here, as it is a central component to your research.

3. Research design

The hypotheses are clearly described and well-motivated by previous research. This is very easy to follow.

However, section 3.2 (model specification and variable setting) needs revision. Firstly, please put each formula on a separate line and number the formula so the reader knows which you are referencing for the basic and moderating models are well. The moderating model is also overlapping within itself. It would also be helpful to have a table of variable names with the corresponding description.

You will see further comments in the text, but please also make it clear why suppliers and customers do not need to be separated when discussing centrality. I assume it is due to knowledge transfer, but this could be more clearly defined to avoid confusion. It may also be helpful to test for the correlation of the variables with each other using a test like Pearson's r.

4. Empirical results and discussion

Please create a table of all the model names and which they are testing. It is very confusing that you introduce the models in the results section. This should rather be in the research design section, so the reader knows to expect results for a number of models, instead of just the three hypotheses. You may want to also introduce topics such as the state vs. non-state owned or life cycle in this section.

It would be good to also present the effect in terms that are easier to interpret. The coefficient and significance tell us one thing, but ideally the reader should also know the strength of the effect in a more practical sense (since the results can also be used with practical implications). You introduce this in section 4.4. (moderating effects) by stating that a 1% rise in digital transformation increases innovation by 0.513% ... etc. This would be helpful for the main model as well. How does innovation performance increase/decrease when centrality is reduced by x%? for example.

Thanks very much for this interesting read!

Reviewer #2: The paper is well structured on the whole; however, it offers us very limited insights or analysis and there are areas for improvement in terms of content.

1. Keywords should ideally be phrases of 2-4 words. Research keywords are not selected

2. The second section (literature review) lacks a detailed review of the model and methodology

what is the other method to analyze the problem?

What is the other methods that used in existing literature?

Why authors use this model to analyze the problem?

3. It is recommended that the authors use charts and diagrams to explain the model and also to analyze the results.

4. Formula needs to be more rigorously expressed. For examples, some symbols are not expressed in italic.

5. The authors need to provide a practical example to prove that the research problem originates from practice and assist readers in understanding this research

6. Some more valuable managerial insights should be provided by sensitive analysis.

7. It is better to compare the results of this study with other similar studies. Also, the benefits of research are described

6. PLOS authors have the option to publish the peer review history of their article (what does this mean?). If published, this will include your full peer review and any attached files.

Reviewer #1: No

Reviewer #2: No

---

## [Author Response · Author response to Decision Letter 0]

13 Oct 2022

Reviewer 1

1. Introduction:

Innovation performance is a central focus of your paper, yet it is not clearly defined for the reader. You state that is is necessary for development and competitiveness, but not what is it or how it could be measured (this is mentioned much later in the methods section). Please clearly define this in the opening paragraph.

Response:

I have revised the definition of innovation performance in the Introduction part and revised the measurement of innovation performance in the Variable Setting part (page2, page6).

2. Literature review:

This section needs a major revision. It is unclear to read, and for some studies you discuss the methodology used, while in others the results of the study as well. Please break this into smaller sections which directly relate to the focus of this research. This could also be broken into sections by results (i.e. studies that found centrality (or some similar characteristic) positively correlated with innovation, and those that found it negatively). You may also want to indicate which are studies based in China, and which are abroad (or a mix of international studies), so that you can discuss the generalizability to other countries/regions.

The reader also does not understand clearly what gap you are filling. From reading the paper it appears the large dataset, as well as the incorporation of the time aspect is where the major contribution lies because you indicate that some other studies also found that centrality may negatively impact innovation. Please also discuss social network theory here, as it is a central component to your research.

Response:

In the updated manuscript, I have carefully redesigned Literature Review. This part is divided into two sections: Supply Chain Concentration and Firms’ Innovation, and Supply Network and Firms’ Development. Some study just found that centrality could positively impact operational performance (Shi et al.,2019). Li et al. (2020) used data from the electronics manufacturing industry of 200 listed firms in China from 2013 to 2017 and discovered that supply network centrality and structural holes were strongly positively connected with innovation performance using a negative binomial model. While we have varied perspectives on their relationship based on big sample data. In the second section, I also have covered social network theory (page 3-4).

3. Research design

The hypotheses are clearly described and well-motivated by previous research. This is very easy to follow. However, section 3.2 (model specification and variable setting) needs revision. Firstly, please put each formula on a separate line and number the formula so the reader knows which you are referencing for the basic and moderating models are well. The moderating model is also overlapping within itself. It would also be helpful to have a table of variable names with the corresponding description.You will see further comments in the text, but please also make it clear why suppliers and customers do not need to be separated when discussing centrality. I assume it is due to knowledge transfer, but this could be more clearly defined to avoid confusion. It may also be helpful to test for the correlation of the variables with each other using a test like Pearson's r.

Response:

I rewrite this part and formed Variable Setting and Model Specification (page 6-7). I have added a table to interpret variables (page 7) and do the correlation of the variables (page 8). I also have added the reason why suppliers and customers do not need to be separated in the text (page 6).

4. Empirical results and discussion

Please create a table of all the model names and which they are testing. It is very confusing that you introduce the models in the results section. This should rather be in the research design section, so the reader knows to expect results for a number of models, instead of just the three hypotheses. You may want to also introduce topics such as the state vs. non-state owned or life cycle in this section.

It would be good to also present the effect in terms that are easier to interpret. The coefficient and significance tell us one thing, but ideally the reader should also know the strength of the effect in a more practical sense (since the results can also be used with practical implications). You introduce this in section 4.4. (moderating effects) by stating that a 1% rise in digital transformation increases innovation by 0.513% ... etc. This would be helpful for the main model as well. How does innovation performance increase/decrease when centrality is reduced by x%? for example.

Response:

I have revised models in Model Specification (page 7). Table 4 (H1and H2) and Table 6 (H3) are the results of basic model and moderating effect model, respectively (page 9,12). As for Moderating Effect Model Test, I rewrite this part with more practical implications (page 11,12).

Reviewer 2

1. Keywords should ideally be phrases of 2-4 words. Research keywords are not selected

Response:

I have selected 4 words as keywords in the revised manuscript (page 1). 

Keywords: network centrality, structural holes, innovation performance, digital transformation

2.The second section (literature review) lacks a detailed review of the model and methodology

what is the other method to analyze the problem?

What is the other methods that used in existing literature?

Why authors use this model to analyze the problem?

Response:

I have restructured Literature Review in the revised manuscript (page 3-4). Prior research has primarily used case studies or questionnaire surveys pertaining to supply chain networks. Some other studies focus on the social network analysis of firms in a sub-sector industry to examine the network-level supply chain relationship. We constructed the basic model and moderating effect model to test the relationship between supply network position and innovation performance and explore the role of digital transformation in the manufacturing field.

3. It is recommended that the authors use charts and diagrams to explain the model and also to analyze the results.

Response:

I have revised 4 Figures for heterogeneity analysis of basic model and discussion of moderating effect model instead of Tables (page 10-12).

4. Formula needs to be more rigorously expressed. For examples, some symbols are not expressed in italic.

Response:

I have checked the formula and revised it by Equation Editor in the Table 1 so that they can be clearly presented (page 7). 

5. The authors need to provide a practical example to prove that the research problem originates from practice and assist readers in understanding this research

Response:

I have revised Introduction part and provided two examples(page 2).

6. Some more valuable managerial insights should be provided by sensitive analysis.

Response:

I have revised the moderating effect analysis and done more practical interpretation(page 11-12).

7. It is better to compare the results of this study with other similar studies. Also, the benefits of research are described

Response:

I have added Comparison with other studies in the text to do comparisons (page 12).

---

## [Decision Letter · Decision Letter 1]

28 Oct 2022

PONE-D-22-18835R1Supply network position, digital transformation and innovation performance: evidence from listed Chinese manufacturing firmsPLOS ONE

Dear Dr. du,

Thank you for submitting your manuscript to PLOS ONE. After careful consideration, we feel that it has merit but does not fully meet PLOS ONE’s publication criteria as it currently stands. Therefore, we invite you to submit a revised version of the manuscript that addresses the points raised during the review process.

We look forward to receiving your revised manuscript.

Kind regards,

Ashkan Memari

Academic Editor

PLOS ONE

Journal Requirements:

Reviewers' comments:

Reviewer's Responses to Questions

**Comments to the Author**

1. If the authors have adequately addressed your comments raised in a previous round of review and you feel that this manuscript is now acceptable for publication, you may indicate that here to bypass the “Comments to the Author” section, enter your conflict of interest statement in the “Confidential to Editor” section, and submit your "Accept" recommendation.

Reviewer #1: All comments have been addressed

Reviewer #2: All comments have been addressed

2. Is the manuscript technically sound, and do the data support the conclusions?

Reviewer #1: Yes

Reviewer #2: Yes

3. Has the statistical analysis been performed appropriately and rigorously? 

Reviewer #1: Yes

Reviewer #2: Yes

4. Have the authors made all data underlying the findings in their manuscript fully available?

Reviewer #1: Yes

Reviewer #2: Yes

5. Is the manuscript presented in an intelligible fashion and written in standard English?

Reviewer #1: Yes

Reviewer #2: Yes

6. Review Comments to the Author

Reviewer #1: Dear Authors,

Thank you very much for your updated manuscript. I have made a few more suggestions in the text, mostly concerning the introduction and literature review section. Here, there are still some parts that are unclear and/or could be improved. Please find my comments in the attached PDF document.

Reviewer #2: (No Response)

7. PLOS authors have the option to publish the peer review history of their article (what does this mean?). If published, this will include your full peer review and any attached files.

Reviewer #1: No

Reviewer #2: No

---

## [Author Response · Author response to Decision Letter 1]

1 Nov 2022

I'd like to thank the reviewer for his/her attentive reading, useful remarks, and constructive ideas, which have substantially enhanced the manuscript's presentation.

Reviewer 1

I have made a few more suggestions in the text, mostly concerning the introduction and literature review section. Here, there are still some parts that are unclear and/or could be improved. 

Response:

I have removed some sentences marked in the Introduction part (page2). 

I have paraphrased some sentences and reorganized the reference order in the Literature Review (page 3-4). 

In the Descriptive Statistics and Analysis part, spearman correlation and multicollinearity analysis are also performed. Even though several values are relative highly correlated; the results reveal that the maximum variance inflation factor (VIF) in Model (1)-(4) is 2.55, 2.54, 1.86 and 1.86, reflecting that there is no multicollinearity between the variables. Some other errs are corrected in the revised copy with track changes.

---

## [Decision Letter · Decision Letter 2]

1 Dec 2022

Supply network position, digital transformation and innovation performance: evidence from listed Chinese manufacturing firms

PONE-D-22-18835R2

Dear Dr. du,

We’re pleased to inform you that your manuscript has been judged scientifically suitable for publication and will be formally accepted for publication once it meets all outstanding technical requirements.

Kind regards,

Ashkan Memari

Academic Editor

PLOS ONE

Additional Editor Comments (optional):

Reviewers' comments:

Reviewer's Responses to Questions

**Comments to the Author**

1. If the authors have adequately addressed your comments raised in a previous round of review and you feel that this manuscript is now acceptable for publication, you may indicate that here to bypass the “Comments to the Author” section, enter your conflict of interest statement in the “Confidential to Editor” section, and submit your "Accept" recommendation.

Reviewer #1: All comments have been addressed

Reviewer #2: All comments have been addressed

2. Is the manuscript technically sound, and do the data support the conclusions?

Reviewer #1: Yes

Reviewer #2: Yes

3. Has the statistical analysis been performed appropriately and rigorously? 

Reviewer #1: Yes

Reviewer #2: Yes

4. Have the authors made all data underlying the findings in their manuscript fully available?

Reviewer #1: Yes

Reviewer #2: Yes

5. Is the manuscript presented in an intelligible fashion and written in standard English?

Reviewer #1: Yes

Reviewer #2: Yes

6. Review Comments to the Author

Reviewer #1: Dear authors,

Thank you very much for your edits. I made a few small grammar suggestions in the first section. This version is very clear and interesting to read!

Reviewer #2: (No Response)

7. PLOS authors have the option to publish the peer review history of their article (what does this mean?). If published, this will include your full peer review and any attached files.

Reviewer #1: No

Reviewer #2: No

---

## [Editor Report · Acceptance letter]

5 Dec 2022

PONE-D-22-18835R2 

Supply network position, digital transformation and innovation performance: evidence from listed Chinese manufacturing firms 

Dear Dr. Du:

I'm pleased to inform you that your manuscript has been deemed suitable for publication in PLOS ONE. Congratulations! Your manuscript is now with our production department. 

Kind regards, 

on behalf of

Dr. Ashkan Memari 

Academic Editor

PLOS ONE